# Analysis of the Impact Resistance of Toecaps by the Finite Element Method: Preliminary Studies

**DOI:** 10.3390/ijerph20010152

**Published:** 2022-12-22

**Authors:** Paulina Kropidłowska, Emilia Irzmańska, Łukasz Gołębiowski, Magdalena Jurczyk-Kowalska, Anna Boczkowska

**Affiliations:** 1Central Institute for Labour Protection-National Research Institute, Wierzbowa 48, 90-133 Lodz, Poland; 2Faculty of Material Science and Engineering, Warsaw University of Technology, 141 Woloska, 02-507 Warsaw, Poland

**Keywords:** protective footwear, toecap impact test, finite element method

## Abstract

A key property in the manufacture of toecaps for protective footwear is resistance to impacts, deformations, and cracking, as the resulting defects may lead to serious workplace accidents involving the lower extremities. The present paper proposes a new approach to qualitative verification of toecap design based on numerical simulations of impact tests. Computational experiments were conducted for toecaps made from different materials (AISI 10450, S235, S355 and A36 steels, as well as Lexan polycarbonate) and characterized by different geometries, which were recreated by 3D scanning. The impact resistance of the toecaps was analyzed using a numerical model simulating an experimental impact test. The results were used to determine the location of critical stresses and to plot equivalent stress maps for the studied toecaps. The finite element analysis of the impact tests was carried out with an explicit elastoplastic finite element code: ANSYS (Ansys, Inc., Canonsburg, PA, USA) with the Explicit Dynamics module of the Workbench solver. The presented analysis of the impact resistance of toecaps by the finite element method for impact simulation may be used to optimize the spatial geometry of toecaps and to verify the construction of toecaps and the material deformations that may occur. In addition, it could eliminate unsuitable materials that are likely to undergo dangerous deformations, and draw attention to the deformation caused by the impact of the toecaps used in footwear in the working environment.

## 1. Introduction

Toecaps are widely used elements of protective footwear that should be designed to meet the requirements of the EU’s Regulation 2016/425 on personal protective equipment [1]. A defining property of toecaps is impact resistance [2], which is evaluated in accordance with the EN 22568 standard [3]. The toecaps applied in protective footwear are usually made of aluminum or high-strength steel [4]. Such materials are also widely used in the automotive industry for clutch disks, bumper reinforcements, and door impact beams to reduce the thickness of those elements, and thus decrease their weight, while preserving good mechanical properties [5,6,7].

The main advantages of metal toecaps include low production costs and good mechanical properties, such as high impact and crack resistance and low deformation, with the disadvantages of being heavy weight and having poor insulation parameters [8,9]. As a result, materials science strives to develop polymeric materials that would be characterized by strength properties and lower weight, compared to steel, while ensuring the appropriate functional properties of toecaps. Recent years have seen an increased application of non-metallic materials including composites incorporating polyester, polyamide, and epoxy resins with nanofillers (carbon, aramid, and glass fibers) [10,11,12,13]. Non-metallic materials are highly attractive due to their flexibility of design and additional features, such as electrical insulation, and very importantly, they provide better thermal insulation than metallic toecaps [14].

Metal toecaps are made from a sheet metal coil stamped in a press to form dome-shaped elements under a weight of approx. 80 tons. The resulting elements are cut in half to obtain a pair of toecaps. In the next step, their bottom edges are folded to create a flange. The toecaps are then hardened by high-temperature treatment and polished [15]. If other toecap metallic materials should be used, a typical production process requires a changeover to adjust machinery to the processing properties of those materials. A quantity of the new material must be purchased and a prototype batch must be produced for its parameters to be evaluated by an external laboratory to ensure compliance with the protective standards. All of these operations entail substantial costs on the part of the manufacturer, in terms of both time and money.

Over the past years, we have seen a new approach to the design and testing of protective footwear and its customization [16,17,18,19,20,21]. Computer simulations are increasingly used to shorten product development time, with the leading technology being analysis by the finite element method optimizing products in terms of weight and strength properties. Such analyses can account for the dynamic phenomena resulting from the impact of free-falling objects. Computer simulation results make it possible to evaluate the protective properties of footwear and conduct further optimization [22,23,24,25].

Optimization of toecap design can be obtained using CAD and CAE tools. Commercial software solutions (for example, ANSYS LS-Dyna or ABAQUS) can be used to simulate the impact behavior of toecaps. Literature data indicate that simulating impact testing has shown good agreement with experimental results. It could provide a better understanding of stress distribution and resultant deformation [26].

According to the available literature data, research is being conducted on the use of high-strength steel to develop the design of the toecaps. Peixinho et al. [27] assessed two prototype steel toecap models the final development of a new light-weight slim toecap for safety footwear. The intended approach of research has been to develop a solution that is based on a greater energy absorption capability due to the formulation of a specific geometric model of toecaps using ultra-high-strength steels, which leads to a substantial weight reduction. The toecap models with significant body-shell thickness reduction to 1.0 mm involve a high ratio of mechanical strength. This type of research is a consequence of the intensive development of other optimized technical solutions with potentiated metal alloys in manufacturing processes.

Currently, the impact resistance of toecaps is evaluated pursuant to the standards EN ISO 20344 (for toecaps placed in protective footwear) and EN ISO 22568 (for the toecaps not inserted in protective footwear), using testing apparatus that should enable the application of appropriate impact energy (expressed in terms of potential energy). Toecaps placed in protective footwear should fulfill the requirement of international standards—EN ISO 20345 (for safety footwear) and EN ISO 20346 (for protective footwear). The difference in the requirements for the toecaps relates to the impact energy against which it is designed to provide protection. For safety footwear, the test is performed with an energy of 200 J, and for protective footwear with an energy of 100 J. Compliance with the requirements of EN ISO 20345 and EN ISO 20346 is verified on the basis of the clearance remaining between the sole and the upper toecap surface after impact.

Laboratory experiments are conducted using an impact tester with a 60 mm long striker made of steel with a Rockwell hardness of 60 HRC, which is positioned parallel to the toecap holder. The two 40 mm long rectangular facets of the impact head form an angle of (90 ± 1)° with the tip rounded at a radius of (3 ± 0.1) mm. The impact tester is equipped with a special mechanism preventing secondary striker impacts to the sample.

The toecap is placed in the tester so that the striker head extends beyond the front and rear edges of the toecap. To determine toecap deflection, a modeling clay cylinder is placed under the rear edge of the toecap (with 2/3 of its diameter inside and 1/3 outside the toecap). During the experiments, a 20 kg striker is dropped on the toecaps from an appropriate height to obtain an impact energy of either (200 ± 4) J or (100 ± 2) J, depending on the protection level claimed by the manufacturer. Impact-induced toecap deflection is assessed based on the smallest deflection point of the modeling clay cylinder, which reflects the clearance under the toecap upon impact. It should be noted that standard tests rely solely on modeling clay cylinder compression without assessing the stress distribution caused by the impact.

The impact test is described in standard BS 7971-4:2002 Protective clothing and equipment for use in violent situations and in training—Limb protectors—Requirements and test methods [28]. During the test, a 20 kg striker falls freely with gravitational acceleration (g = 9.8 m/s^2^) from a height of 1075 mm. Initially, a body in free fall has a kinetic energy of E_ko_ = 0 and gravitational potential energy E_p_ = mgh, whereas at the end of the fall its kinetic energy increases to E_k_ = mv^2^/2, with its potential energy being E_pk_ = 0. In light of the law of conservation of mechanical energy, the initial and final sums of the kinetic and potential energies of a body are equal, which means that:
(1)$$mgh=\frac{mV^{2}}{2}$$

Thus, the speed of the body at the time of impact on the surface can be obtained from the following equation:
(2)$$V=\sqrt{2gh}$$


Since a body in free fall moves with constant gravitational acceleration (g), its final speed can be represented as:*V* = *gt*(3)

The preliminary studies of the analysis of the impact resistance of toecaps by the finite element method described in this paper can provide a qualitative verification of toecap design and help in optimizing toecap geometry and material composition; it can be also used to screen out materials that are unsuitable for protective toecaps in terms of deformations that are dangerous for the user in the working environment.

## 2. Materials and Methods

The preparation of the research material was initiated with 3D scanning of commercially available toecaps (metal and polymeric) and reconstructing the spatial geometry of the toecaps in a CAD file. The next step was to develop a computational model with boundary conditions reflecting the experimental impact test as described in the standard and including the mechanical properties of the selected materials in analysis. The Johnson–Cook model was applied and strength analysis and qualitative analysis of the selected geometries was evaluated.

The developed computational model simulates impact testing of steel and polycarbonate toecaps. Numerical strength modeling revealed critical stress areas in the studied toecap types.

### 2.1. Geometry Preparation Methodology

The study involved toecaps of two geometries (Table 1). Toecaps with geometry A were made from a thermoplastic polymer (polycarbonate), while type B toecaps were made from different grades of steel. The actual geometry of the toecaps was reconstructed by means of 3D scanning (Figure 1). Toecap surface was mapped with a cloud of points used to create a triangular mesh, and finally a CAD model. Toecap surface was scanned using structural white LED light with an accuracy of 0.04 mm and a resolution of 73 points/mm^2^.

### 2.2. Johnson–Cook Model and Material Selection

Owing to incomplete material identification of the scanned toecaps, five materials were selected based on literature data, a polymer and four grades of steel:Lexan—a polycarbonate;AISI 10450—a quality heat-treatable non-alloy steel;S235—a non-alloy structural steel;S355—a high-strength low-alloy structural steel;A36—a high-strength low-alloy steel.

Numerical calculations were based on the basic mechanical properties of the selected materials provided in Table 2. Simulations also took into account the Johnson–Cook parameters describing material hardening upon impact. The model is given by the multiplicative form of the constitutive equation, which is a function of strain, strain rate, and temperature:
(4)$$\sigma (\varepsilon ,\dot{\varepsilon },T)=\left(A+B\varepsilon^{n}\right)\left(1+C\ln\left(\frac{{\dot{\varepsilon }}_{p}}{{\dot{\varepsilon }}_{0}}\right)\right){\left(1-\left(\frac{T-T_{R}}{T_{m}-T_{R}}\right)\right)}^{m}$$
where:

*A* = initial static yield stress for the reference parameters: ambient temperature T_R_ and strain rate ε’_0_;

*B* = strain hardening constant;

*n* = strain hardening exponent;

*m* = thermal softening exponent;

*C* = strain rate hardening coefficient;

*T* = sample temperature;

*T_m_* = melting point. 

The material constants A and B, as well as n, describe hardening under quasi-static conditions; the C parameter was determined by an interaction process from tension test data and it can be also found in Table 2. ε_p_ is the accumulated plastic strain and ε_0_ is the reference strain rate which, in this work, was taken as 1 s^−1^. The effects of temperature were not included, as numerical analyses were performed for the standard temperature and heat values concerning to plastic strain are far from influence at these strain rates under consideration. The empirically determined Johnson–Cook constants for the studied materials are given in Table 3. 

## 3. Analysis of the Impact Resistance of Toecaps by the Finite Element Method

### Finite Element Modeling

Mechanical strength analyses were conducted using numerical toecap models reflecting experimental impact attenuation tests pursuant to the standard BS 7971-4: 2002. Finite element modeling (FEM) was implemented in ANSYS Explicit Dynamics software. The computational model incorporated geometrical models, material properties, and boundary conditions. 

Analyses included the toecap geometry types A and B, which were reconstructed by 3D scanning of actual toecaps. The computational model was divided into four-node elements in a 3D space.

The numerical model was performed with three independent parts: the representative striker body, a bottom layer for the constrained support of the toecap, and the toecap model which performed the numerical analysis. The analysis model was developed on 3D solid elements. The toe cap model was discretized into tetrahedral structural solid elements with sizing mesh control. The other two solid parts were discretized into quadrangle-based prism elements (Figure 2).

The FEM boundary conditions included: the striker velocity vector immediately before impact V_k_ = 4480 mm/s; loading Q = 18.8 kg applied to the upper surface of the striker (to give an overall loading of 20 kg including the weight of the striker of 1.2 kg); and attachment with three blocked degrees of freedom (x, y, z) on the upper surface of the baseplate as shown in Figure 3.

## 4. Results and Analysis

Analysis of the impact resistance of toecaps by the finite element method showed that the maximum equivalent stress for one of the four steel toecap models exceeded the strength limits for the material (AISI steel), leading to toecap wall failure and fracturing. The obtained stress values were within acceptable limits for toecaps made from high-strength structural steel S355 and A36 (Table 4).

Results for the two selected toecap models (geometry type A made from Lexan polycarbonate and type B from S235 steel) are presented in the form of Huber–von Mises equivalent stress and deformation maps. The deformation is described by the node with the maximum displacement in the Z direction, this represents how much the toecap is compressed.

In the analyzed models, the location of maximum stress and deformation largely depended on their shape. In polycarbonate toecaps type B, the greatest equivalent stress (80.3 MPa) was found in the middle of the upper wall near the area of impact (Figure 4), with the resulting deformation being 7.4 mm (Figure 5). In turn, in steel toecaps type A the highest equivalent stress (approx. 500 MPa) occurred on the topmost edge of the upper wall and in the bottom segment near the baseplate (Figure 6). In toecaps made from S235 steel, upper wall deformation amounted to 5.34 mm (Figure 7).

Analysis of results also shows the importance of toecap material in safety footwear. Figure 8 and Figure 9 show the node with the highest stress and deformation-value-over-time charts. Whereas all steel toecaps were characterized by the same geometry (type A), the strength curve for AISI 1045 steel differed unfavorably from the others, exhibiting a highly irregular course. Indeed, toecap behavior under dynamic loads depends on the type of steel used. Ultimately, the stress–strain state of toecaps is affected by the material hardening phenomena caused by striker impact. In polycarbonate toecaps, there was a local stress concentration around the area of impact, whereas in steel toecaps the stress was distributed over a considerably larger area. 

## 5. Discussion

The literature predominantly focuses on the static aspects of protective product testing, with the evaluated factors being correlations with absorption energy, mean crushing load, maximum peak load, stress distribution, and toecap deformation [5,12,34,35].

Some authors have explored cutting-edge technologies (mainly numerical modeling) in determining stress distribution, as well as toecap thickness and geometry optimization, with a view to improving resistance to mechanical factors [4,13,34,36]; these studies conducted computer simulations of the impact resistance of toecaps made from 1.2–1.8 mm thick martensitic steel sheets (Mart1200) using ANSYS software. Experimental tests were conducted using a Pegasil E-99 apparatus with a 25 kg striker and recorded with a high-speed camera at 5000 fps. The data were processed using TEMA Motion software. The numerical model consisted of the striker, toecap, and baseplate (support for the toecap). A modification of toecap geometry was found to improve its strength properties. Ribbed toecaps made of 1.2 mm thick steel revealed greater impact resistance as compared to non-ribbed toecaps made of steel sheets that were thicker by 0.3 mm.

An understanding of damage mechanisms at high stress and strain values is critical to the design of structures exposed to random impacts. Costa et al. [4] evaluated the impact resistance of steel toecaps made of two types of high-strength fully martensitic steels with a tensile strength of 1200 and 1400 MPa. The objective was to examine different toecap geometries with a view to maximizing the mechanical properties of toecaps while maintaining the ductility levels of steel needed for production processes. It was found that toecaps with considerably reduced thickness also revealed high impact energy absorption, despite significant deformation levels. In the author’s studies the obtained stress values were within acceptable limits for toecaps made from high-strength structural steel S355 and A36.

Rodrigues et al. [37] evaluated a solid mechanics toolbox, built in the open-source computational library, OpenFOAM, to simulate compression and impact tests, which was used to assess commercially available plastic toecaps. The stresses developed during the impact tests are distinct from those in the compression tests, exhibiting a wave-like propagation form from the top to the bottom. In the present study, in polycarbonate toecaps, there was a local stress concentration around the area of impact. The greatest equivalent stress was found in the middle of the upper wall near the area of impact, with the resulting deformation being 7.4 mm (Figure 5).

Lee et al. [13] analyzed the strength properties of fiber-reinforced plastic (FRP) toecaps, made from a glass fiber polyester composite, as compared to steel toecaps. ABAQUS software was used to conduct computer simulations of the dynamic and static forces acting on toecaps to optimize their thickness. The results of computer simulations were verified experimentally. In the case of steel toecaps, FEM revealed high tensile stress in their front part and compressive stress in the upper part, causing plastic deformation. In the case of composite toecaps, the highest compressive and tensile stress was found in the front part, which affected the type of impact-induced damage. The upper part of the toecaps revealed delamination and glass-fiber breakage. Preliminary analysis indicated the same values of toecap deformation at a wall thickness of 3.2 mm and 1.7 mm for composite and steel toecaps, respectively. The study also addressed the effect of reinforcement parameters (fiber type and distribution in the matrix) on the strength properties of the obtained composites. Toecaps containing layers of glass fibers with +45°/−45° orientation were characterized by the lowest static compression damage. In other orientations, the fibers tended to break. The application of a 0° orientation led to the highest maximum deformation. In the other cases, deformation values were similar to, or lower than, those observed for the steel toecaps. Impact tests revealed that the highest stress and strain was lower in the composite toecaps than the steel ones. The higher absorbed energy was associated with breakage of the reinforcement and the matrix during the test. The composite toecaps were also characterized by much lower permanent strain. In terms of impact and compression resistance, the best results were found for the composite with glass fibers oriented at +45°/−45° and with glass mat layers on the external surface. Following damage to the upper part of the toecap, the load was transferred to a greater extent to its front part. In the present study, in polycarbonate toecaps (type B), the greatest equivalent stress (80.3 MPa) was found in the middle of the upper wall near the area of impact (Figure 4).

The available literature describes numerical simulations utilizing the finite element method to determine the highest stress concentrations and optimize material thickness and geometry, with a view to improving the impact resistance of toecaps. 

The presented computational impact test analysis could be a useful support to the process of toecap design to increase user safety and comfort. Furthermore, it can be used to optimize the spatial geometry of toecaps and to verify the construction of toecaps and the material deformations, which pose a direct hazard to safety footwear users, that may occur.

## 6. Conclusions

The objective of the study was to perform analysis of the impact resistance of toecaps using the finite element method as a preliminary study for further, in-depth research into the analysis of the impact resistance of toecaps. The standard approach in toecap manufacturing is to produce a trial batch for experimental tests and verification of their impact resistance, which is both costly and time consuming. The numerical analysis of the toecap testing makes it possible to conduct preliminary evaluation without purchasing the necessary materials and launching production. 

This study presented results on the response of safety toecap models, made of various steels and Lexan, investigated under quasi-static conditions at impact strain rates. The undertaken work contributes to better understanding the structural impact behavior of toe cap models. 

The computational experiments were conducted on toecaps made from different materials (AISI 10450, S235, S355 and A36 steels, as well as Lexan polycarbonate) and characterized by different geometries. The results determined the location of critical stresses and plotted equivalent stress maps for the toecaps. In the analyzed toecap models, the location of maximum stress and strain largely depended on their shape and material. In polycarbonate toecaps the greatest equivalent stress was found in the middle of the upper wall near the area of impact. On the other hand, in steel toecaps, the highest equivalent stress occurred on the top-most edge of the upper wall and in the bottom segment near the baseplate. 

Numerical simulation of mechanical strength behavior also provides opportunities for designing and optimizing toecaps made from composite materials, such as FRP. The application of material modeling at the stage of toecap design may accelerate the implementation of new materials with improved protective properties.

## Figures and Tables

**Figure 1 ijerph-20-00152-f001:**
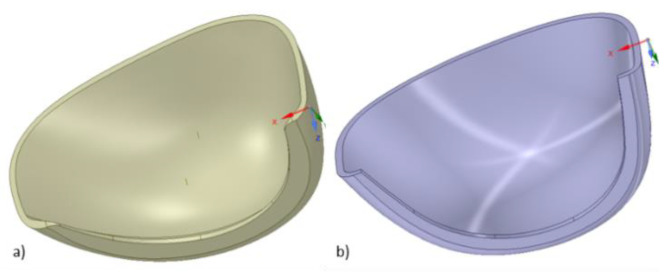
Toecap geometries reconstructed by means of 3D scanning: (**a**) type A—polymeric toecaps, (**b**) type B—steel toecaps.

**Figure 2 ijerph-20-00152-f002:**
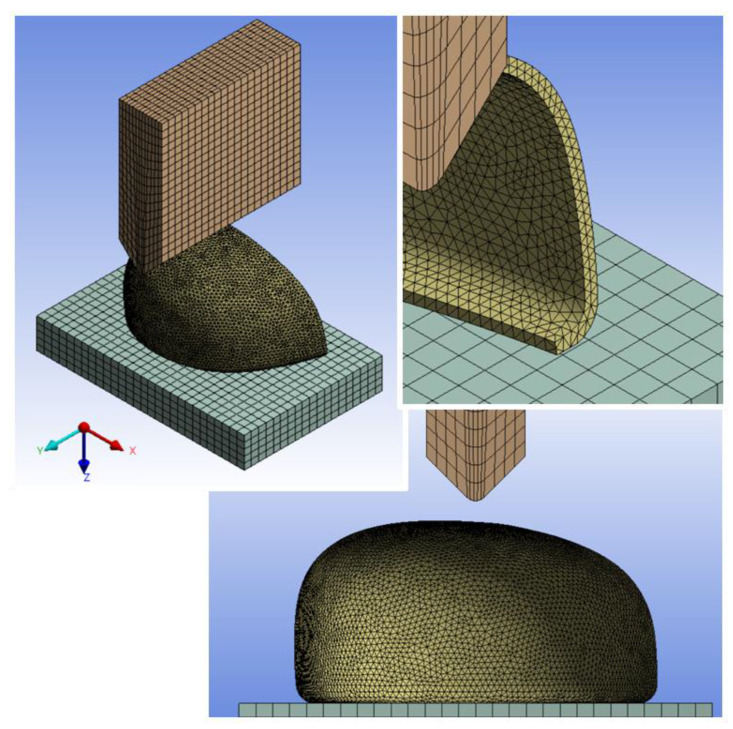
Finite element mesh for toecap type A (polymeric), striker, and table.

**Figure 3 ijerph-20-00152-f003:**
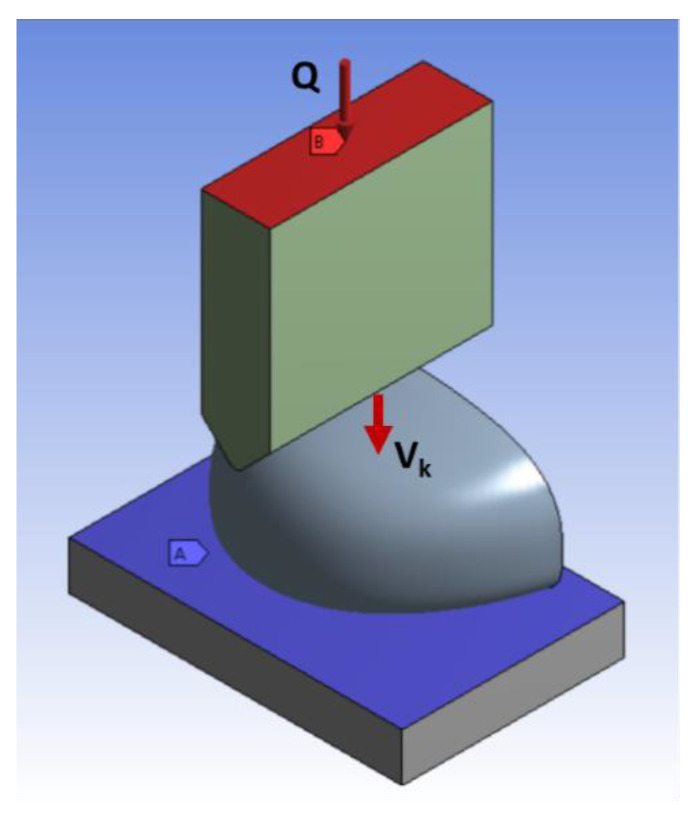
Boundary conditions for toecap FEM.

**Figure 4 ijerph-20-00152-f004:**
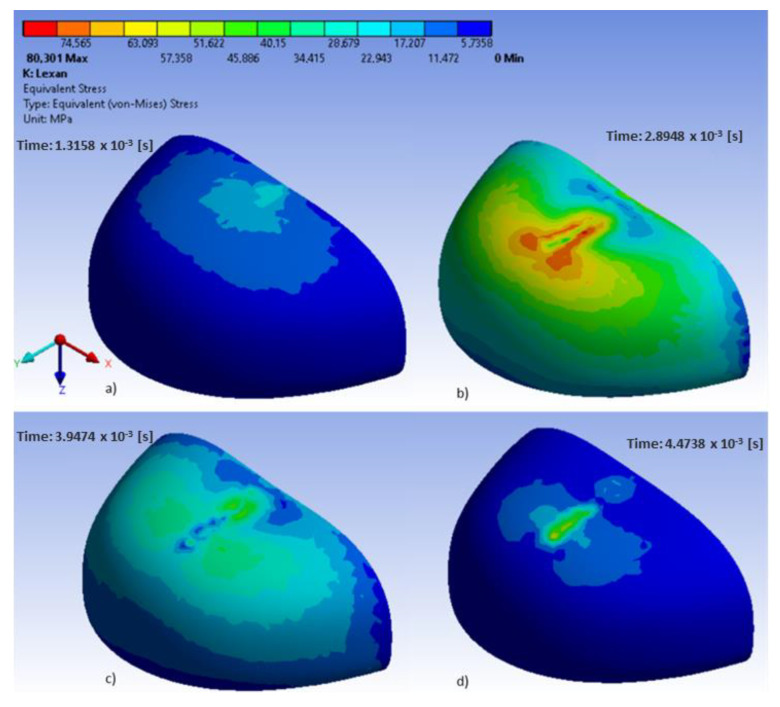
Equivalent stress in polycarbonate toecaps at different time steps: (**a**–**d**) external side.

**Figure 5 ijerph-20-00152-f005:**
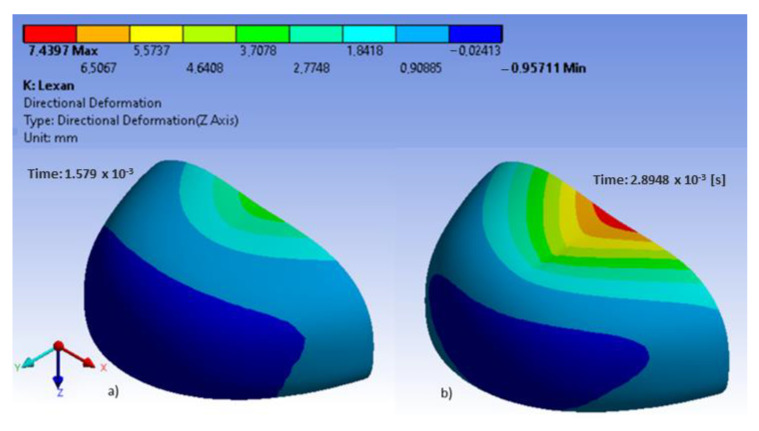
Deformation (approx. 7.5 mm) of polycarbonate toecaps in two time steps: (**a**,**b**) external side.

**Figure 6 ijerph-20-00152-f006:**
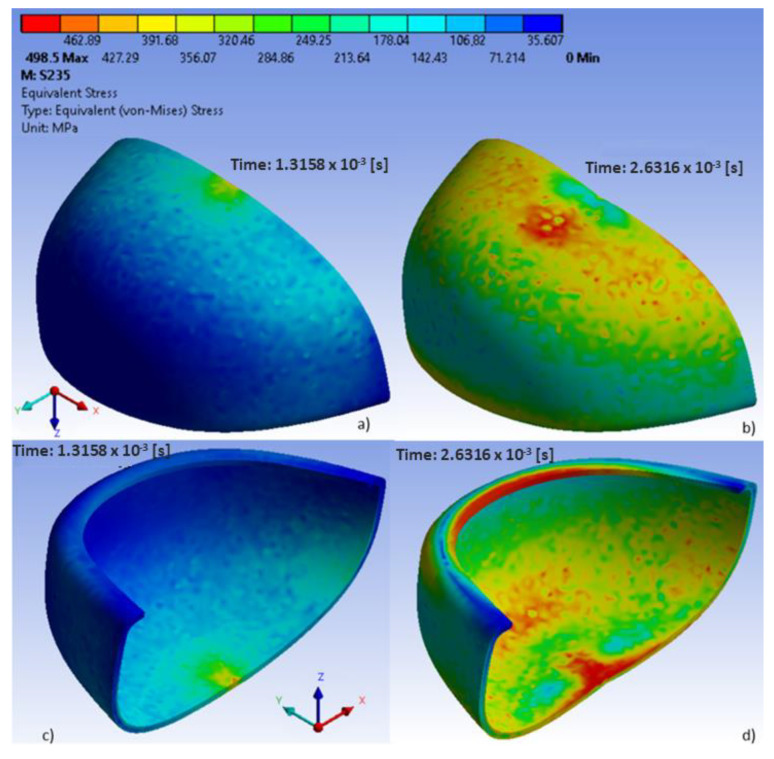
Equivalent stress in steel toecaps at different time steps: (**a**,**b**) external surface, (**c**,**d**) internal surface.

**Figure 7 ijerph-20-00152-f007:**
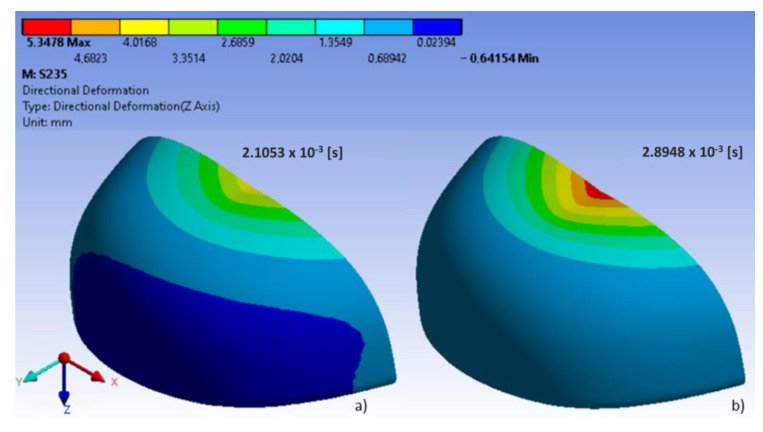
Deformation (approx. 5.3 mm) of steel S235 toecaps in two time steps: (**a**,**b**) external side.

**Figure 8 ijerph-20-00152-f008:**
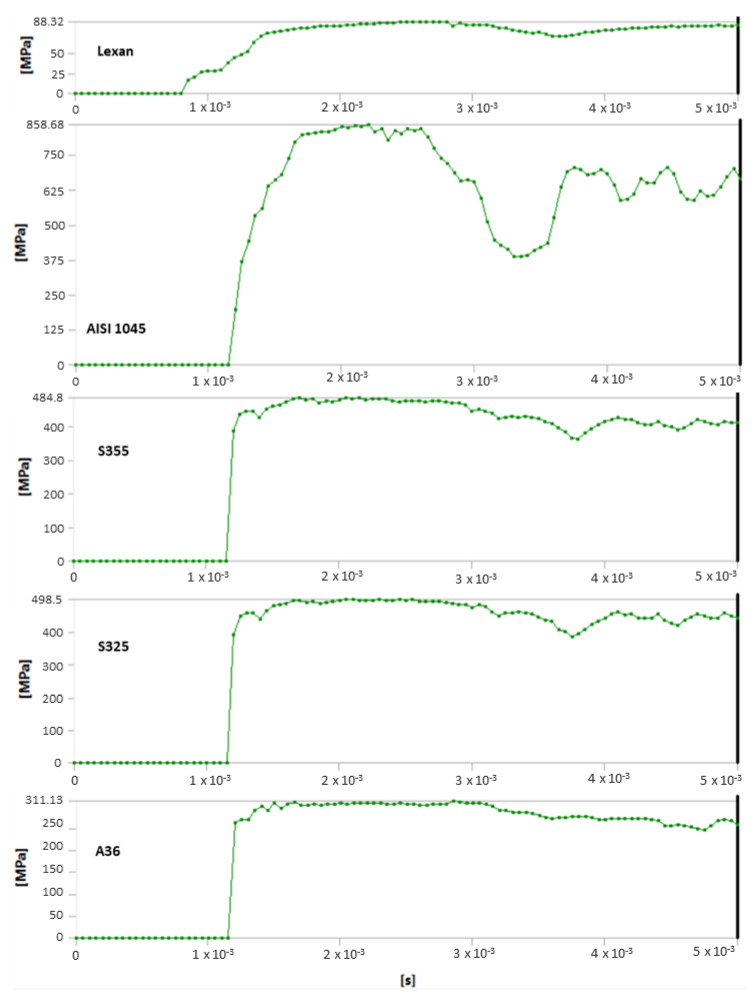
Huber–von Mises equivalent stress-over-time diagrams for the studied models.

**Figure 9 ijerph-20-00152-f009:**
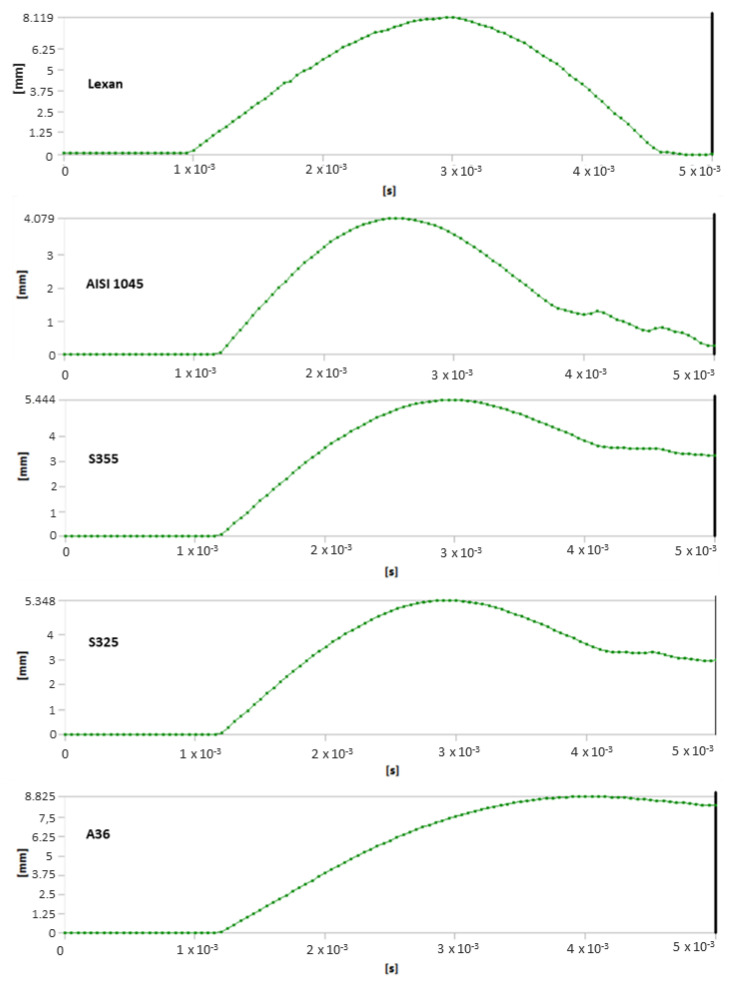
Deformation-over-time diagrams for the studied models.

**Table 1 ijerph-20-00152-t001:** Tested toecaps.

Toecap Type	Mean Thickness [mm]	Photograph	Projection Generated by 3D Scanning
Type A	2.2	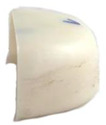	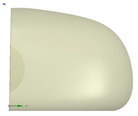 side view
		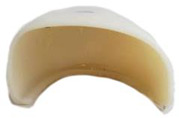	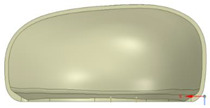 rear view
Type B	2.8	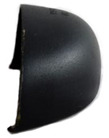	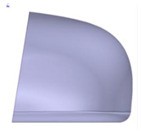 side view
		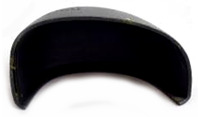	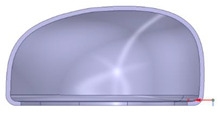 rear view

**Table 2 ijerph-20-00152-t002:** Physical and mechanical properties of the studied materials.

Material	Density (g/cm^3^)	Young’s Modulus (GPa)	Poisson Coefficient (–)	Tensile Strength (MPa)
Lexan [29]	1.19	2.54	0.34	–
AISI 1045 [30]	7.85	200	0.30	560–850
S235 [31]	7.85	200	0.30	360–510
S355 [32]	7.85	210	0.30	510–680
A36 [33]	7.80	20.0	0.26	490–620

**Table 3 ijerph-20-00152-t003:** Johnson–Cook constants for the studied materials.

Material	*A* Yield Stress (MPa)	*B* Hardening Constant (MPa)	*n* Hardening Exponent (–)	*C* Rate Constant (–)
Lexan [29]	75.8	68.9	1	0
AISI 1045 [30]	553.0	600.0	0.234	0.0134
S235 [31]	480.0	153.0	0.360	0.0141
S355 [32]	448.0	782.0	0.562	0.0247
A36 [33]	286.1	500.1	0.228	0.0220

**Table 4 ijerph-20-00152-t004:** Impact test results for the studied toecap types in relation to the tensile strength of the materials used.

Material	Deformation (Displacement in Z Direction) (mm)	H–vM Equivalent Stress (MPa)	Tensile Strength (MPa)
Lexan	7.400	80.3	Not applicable
AISI 1045	4.048	858.0	560–850 *
S235	5.340	494.0	360–510 *
S355	5.440	484.0	510–680 *
A36	6.520	311.0	490–620 *

* Tensile strength ranges defined in the applicable standards for the steels used.

## Data Availability

Not applicable.

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
