# Peer review of "Analysis of the Impact Resistance of Toecaps by the Finite Element Method: Preliminary Studies"

_ijerph, 2022, doi:10.3390/ijerph20010152_

Round 1

Reviewer 1 Report (Previous Reviewer 3)

No further comments - thank you for acknowledging or addressing previous comments.

Author Response

I am attaching a comment.

Emilia Irzmanska

Reviewer 2 Report (New Reviewer)

English should be improved on Section 1 - Introduction.

1) Minor Spell check is required:

Abstract: "The present paper proposes a new approach to for qualitative verification toecap design" ==> "The present paper proposes a new approach to qualitatively verify the toecap's design" 

Line 57: a dot is missing before "Computer..."

Line 69: Please clarify the impact energy difference "after a 100 J (occupational footwear) or 200 J impact (safety footwear)."

Line 81: "which the clay cylinder was compressed"

2) Comments and recommendations

·       The introduction is poor regarding the state of the art. There are several recent articles where the impact and compressive behavior of metallic and non-metallic toecaps are accessed using FVM and FEM approaches. Some examples that should be on the introduction and discussion sessions:

https://doi.org/10.3390/polym13244332%EF%BB%BF

https://doi.org/10.1590/1980-5373-MR-2022-0167

·       Also, it lacks justification for the novelty of this work, regarding the actual state of the art in this field of research. It does not explain what new fiding this article will bring, compared to the bibliography.

·       Regarding Table 1, are the authors sure that the Type A toecap is made of Polycarbonate? It looks like one made of polyamide or thermoset material.  It seems that the authors used a commercial plastic toecap to scan the geometry, and not a specific polycarbonate toecap. Please, clarify this and correct the text in the article (line 120).

·       On “Materials and methods” it should be described how the mesh was generated (for the striker and for the toecap). Also, the restrains/boundary conditions of the simulation setup, so anyone could replicate it. If the authors used conditions from the literature, it must the referenced.

·       Line 216 – 219: The authors state “The adopted research procedure and simulations indicate that stress and strain values can be minimized by the application of an appropriate material or by a variety of design solutions, such as increasing toecap thickness in certain regions, and designing geometry (…). But the adopted research only shows how stress distribution is affected by the type of material and geometry. There is no study on this article that correlated the difference in toecap thickness or design on the mechanical behavior under impact conditions. Please revise this phase.

·       Table 4: The deformation results is the displacement in the ZZ direction? In other words, this represents how much the toecap is compressed? If so, it should be explicit which point you used to determine this value (probably this value is to simulate how a real clay cylinder would compress).

·       Figure 8 and Figure 9: which cell are you using to plot those results? It should be stated if you choose a point, or you choose the cell which has the highest stress value during the simulation.

·      Section 5 – Discussion: This should be in the Introduction section… The authors do not discuss anything in Section 5. They only describe what are in the literature… Please, add more references to the literature review (https://doi.org/10.3390/polym13244332%EF%BB%BF; https://doi.org/10.1590/1980-5373-MR-2022-0167).

·      Section Conclusions (please, correct the section number): It lacks a conclusion regarding the findings of these work. It should state the main conclusion and the its importance for the knowledge that is being created.

Author Response

Responses to Reviewers concerning Manuscript ID: ijerph- 2065617

We would like to thank the Reviewer for comments, which have enabled us to improve our paper both in its content and presentation. All revisions have been marked in color: red

1) Minor Spell check is required:

Abstract: "The present paper proposes a new approach to for qualitative verification toecap design" ==> "The present paper proposes a new approach to qualitatively verify the toecap's design" 

corrected

Line 57: a dot is missing before "Computer..."

corrected

Line 69: Please clarify the impact energy difference "after a 100 J (occupational footwear) or 200 J impact (safety footwear)."

Toecaps placed in the protective footwear shall fulfill the requirement of international standards - EN ISO 20345 (for safety footwear) and EN ISO 20346 (for protective footwear). The difference in the requirements for the toecaps relates to the impact energy against which it is designed to provide protection. For protective footwear, the test is performed with an energy of 200 J, and for protective footwear with an energy of 100 J. Compliance with the requirements of EN ISO 20345 and EN ISO 20346 is verified on the basis of the clearance remaining between the sole and the upper toecap surface after impact.

Line 81: "which the clay cylinder was compressed"

Impact-induced toecap deflection is assessed based on the smallest deflection point of the modeling clay cylinder which reflects the clearance under the toecap upon impact. It should be noted that standard tests rely solely on modeling clay cylinder compression without assessing the stress distribution caused by the impact.

2) Comments and recommendations

  • The introduction is poor regarding the state of the art. There are several recent articles where the impact and compressive behavior of metallic and non-metallic toecaps are accessed using FVM and FEM approaches. Some examples that should be on the introduction and discussion sessions:

Corrected

  • Also, it lacks justification for the novelty of this work, regarding the actual state of the art in this field of research. It does not explain what new fiding this article will bring, compared to the bibliography.

The novelty of this work,  comes from the  provide a better understanding of stress distribution and resultant deformation over time by different geometries tecap.

  • Regarding Table 1, are the authors sure that the Type A toecap is made of Polycarbonate? It looks like one made of polyamide or thermoset material.  It seems that the authors used a commercial plastic toecap to scan the geometry, and not a specific polycarbonate toecap. Please, clarify this and correct the text in the article (line 120).

Polycarbonate toecap was used in the study, which can be confirmed by FTIR-ATR studies that were performed in connection with another research thread the authors were pursuing. A graph of the FTIR-ATR spectrum is included for confirmation.

Fig. FTIR-ATR spectrum of toecap material

  • On “Materials and methods” it should be described how the mesh was generated (for the striker and for the toecap). Also, the restrains/boundary conditions of the simulation setup, so anyone could replicate it. If the authors used conditions from the literature, it must the referenced.

The numerical model was performed by three independent parts: the representative striker body, a bottom layer for the constrained support of the toe cap, and the toe cap model which performed the numerical analysis.

The analysis models was developed on a 3D solid elements. The toe cap model was discretized into tetrahedral structural solid elements with sizing mesh control. The other two solid parts was discretized into quadrangle-based prism elements.

  • Line 216 – 219: The authors state “The adopted research procedure and simulations indicate that stress and strain values can be minimized by the application of an appropriate material or by a variety of design solutions, such as increasing toecap thickness in certain regions, and designing geometry (…). But the adopted research only shows how stress distribution is affected by the type of material and geometry. There is no study on this article that correlated the difference in toecap thickness or design on the mechanical behavior under impact conditions. Please revise this phase.

Corrected

  • Table 4: The deformation results is the displacement in the ZZ direction? In other words, this represents how much the toecap is compressed? If so, it should be explicit which point you used to determine this value (probably this value is to simulate how a real clay cylinder would compress).

The deformation is described by the node the node with the maximum displacement  in the Z direction over time, this represents how much the toecap is compressed.

Corrected in table 4.

  • Figure 8 and Figure 9: which cell are you using to plot those results? It should be stated if you choose a point, or you choose the cell which has the highest stress value during the simulation.

Fig. 8-9 show the node with the highest stress and deformation value charts during the simulation over time.

  • Section 5 – Discussion: This should be in the Introduction section… The authors do not discuss anything in Section 5. They only describe what are in the literature… Please, add more references to the literature review (https://doi.org/10.3390/polym13244332%EF%BB%BF; https://doi.org/10.1590/1980-5373-MR-2022-0167).

Corrected

  • Section Conclusions (please, correct the section number): It lacks a conclusion regarding the findings of these work. It should state the main conclusion and the its importance for the knowledge that is being created.

The computational experiments were conducted for toecaps made from different ma-terials (AISI 10450, S235, S355 and A36 steels as well as Lexan polycarbonate) and char-acterized by different geometries. The results obtained to determine the location of critical stresses and to plot equivalent stress maps for the toecaps. In the analyzed toecaps models, the location of maximum stress and deformation largely depended on their shape and mate-rial. In polycarbonate toecaps the greatest equivalent stress was found in the middle of the upper wall near the area of impact. In turn, in steel toecaps, the highest equivalent stress occurred on the top-most edge of the upper wall and in the bottom segment near the baseplate.

Round 2

Reviewer 2 Report (New Reviewer)

I would like to thank the authors for responding to all comments and suggestions. 

Just two minor revisions:

Table 4: correct the word "dispalcement" ==> displacement 

line 308: correct the word "asses" ==> assess

Best of luck,

Author Response

Responses to Reviewers concerning Manuscript ID: ijerph- 2065617

We would like to thank the Reviewer for comments, which have enabled us to improve our paper both in its content and presentation. All revisions have been marked in color: red

Comments and Suggestions for Authors

I would like to thank the authors for responding to all comments and suggestions.

Just two minor revisions:

Table 4: correct the word "dispalcement" ==> displacement

corrected

line 308: correct the word "asses" ==> assess

corrected

This manuscript is a resubmission of an earlier submission. The following is a list of the peer review reports and author responses from that submission.

Round 1

Reviewer 1 Report

- Overall -

I hope that my comments below are informative for future revision of the paper. Sorry, I cannot recommend the publication of the current version of the manuscript because of the followings.

- Abstract and Conclusion -

The purpose (objective, goal) and subsequent finding (conclusion) are indeed unclear

[L17] In spite of the claim of the authors, ‘numerical tool’ was not provided in the paper. The authors simply carried out finite element analysis (FEA) using a commercial FEA package.

[L18-19] In spite of the claim of the authors, ‘spatial geometry’ and ‘appropriate material’ were not selected in the paper at all. The readers of this paper cannot receive any assistance from this paper in doing the selection.

[L304-305] In spite of the claim of the authors, ‘directions for geometry’ and ‘material optimization’ was not carried out at all in this paper. The readers of this paper cannot receive any assistance from this paper in obtaining the forgoing solutions.

[L305-306] ‘In the case of steel toecaps, it is necessary either to use high-strength steel grades or increase wall thickness.’ -> It cannot be the finding or conclusion of this paper. People already know this point as general knowledge before reading this paper. 

- Main text -

[L43] ’80 tons’ cannot be the pressure.

[L73] ’20,000 g’ -> 20 kg

[L142] ‘with the heat values associated with plastic deformation being much higher than those 142 generated at the strain rates considered’ -> If then, stress relaxation (thermal softening of flow stress) due to heat must be calculated and the strain rate effect must be neglected. It is a kind of self-contradiction.

[Table 2] Young’s modulus value cannot be such values. If correctly written, FEA was not carried out correctly.

[Fig 5 and 9] Unnecessary.

[Fig 11] Must be plotted all together in a single diagram. Plus Fig 6e and 9e must be eliminated as they are repeating Fig 11.

[new Fig] Deformation vs time profiles of investigated materials need to be plotted all together in a single diagram like the modified Fig 11

[Other figures] The number of stress contour figures of materials should be minimized and they should be in a single Figure nuber for all investigated materials.

[Serious] The strain rate should be characterized in FEA and useful conclusion should be extracted in impact analysis. Otherwise, no reason is found for using the Johnson Cook model. If the strain rate is elucidated, it may not be high; it may reveal that the strain rate effect can be neglected in FEA. Even though such conclusion is drawn out, it is useful for people in that this paper prevent from wrong direction in toecap design. If the necessity of considering the strain rate effect is suggested by this paper, it is also useful for people in that this paper let people go to right direction.

Author Response

Reviewer #1:

Thank you for your review and comments. Changes have been made in the body of the text.

Below please find our specific responses.

The purpose (objective, goal) and subsequent finding (conclusion) are indeed unclear

Changes have been made to the abstract and conclusion. In addition, a title change was proposed: Analysis of the impact resistance of toecaps by the finite element method - preliminary studies, and added co-author prof. Anna Boczkowska.

 [L17] In spite of the claim of the authors, ‘numerical tool’ was not provided in the paper. The authors simply carried out finite element analysis (FEA) using a commercial FEA package.

Corrected and supplemented the description: The finite element analysis of the impact tests was carried out with an explicit elastoplastic finite element code: ANSYS (Ansys, Inc.) with the Explicit Dynamics module of the Workbench solver.

[L18-19] In spite of the claim of the authors, ‘spatial geometry’ and ‘appropriate material’ were not selected in the paper at all. The readers of this paper cannot receive any assistance from this paper in doing the selection.

Corrected and supplemented the description: The presented numerical tool for impact simulation may be used by the designers of safety footwear to verify the construction of toecaps

[L304-305] In spite of the claim of the authors, ‘directions for geometry’ and ‘material optimization’ was not carried out at all in this paper. The readers of this paper cannot receive any assistance from this paper in obtaining the forgoing solutions.

Corrected and supplemented the description: This study presented results on the dynamic response of safety toecap models made of various steels and lexan.

 [L305-306] ‘In the case of steel toecaps, it is necessary either to use high-strength steel grades or increase wall thickness.’ -> It cannot be the finding or conclusion of this paper. People already know this point as general knowledge before reading this paper. 

Corrected and supplemented the description: The undertaken work contributes to better understand the structural impact behavior of toe cap models. Also it is useful for people in that this paper prevent from wrong direction in toecap design

- Main text -

[L43] ’80 tons’ cannot be the pressure.

Corrected

 [L73] ’20,000 g’ -> 20 kg

Corrected

 [L142] ‘with the heat values associated with plastic deformation being much higher than those 142 generated at the strain rates considered’ -> If then, stress relaxation (thermal softening of flow stress) due to heat must be calculated and the strain rate effect must be neglected. It is a kind of self-contradiction.

Corrected:  The effects of temperature were not included as numerical analyses were performed for the standard temperature and heat values concerning to plastic strain are far from influence at these strain rates under consideration

[Table 2] Young’s modulus value cannot be such values. If correctly written, FEA was not carried out correctly.

Corrected.

[Fig 5 and 9] Unnecessary.

Figures 5 and 8 were deleted.

[Fig 11] Must be plotted all together in a single diagram. Plus Fig 6e and 9e must be eliminated as they are repeating Fig 11.

Graphics 6e and 9e have been removed from the manuscript.

[new Fig] Deformation vs time profiles of investigated materials need to be plotted all together in a single diagram like the modified Fig 11

New figure - a summary of deformation characteristics for all analyzed models has been added to the manuscript

[Other figures] The number of stress contour figures of materials should be minimized and they should be in a single Figure nuber for all investigated materials.

The fonts for the legend describing the stresses were generated automatically by the calculation program, hence the lack of editing capabilities.

[Serious] The strain rate should be characterized in FEA and useful conclusion should be extracted in impact analysis. Otherwise, no reason is found for using the Johnson Cook model. If the strain rate is elucidated, it may not be high; it may reveal that the strain rate effect can be neglected in FEA. Even though such conclusion is drawn out, it is useful for people in that this paper prevent from wrong direction in toecap design. If the necessity of considering the strain rate effect is suggested by this paper, it is also useful for people in that this paper let people go to right direction.

We would like to thank the Reviewer for this comment. The material constants A and B as well as n describe hardening under quasi-static conditions, In this case, it was picked a strain rate of 1 s−1 to improve application conditions. The parameter C refers to dynamic conditions. For all materials have performed the same predictions for the Johnson–Cook equation. It is also possible to observe that this constitutive equation might represent more difficulty to cover the entire strain-rate rage. The original form of the Johnson–Cook equation is normally limited to higher strain rates up to 104 s−1, however it shows greatest interest here with a lower approximation to the quasi-static behavior for strain rates.

Reviewer 2 Report

Manuscript Number: ijerph-1878045

Manuscript title: Numerical simulation of toecap impact test in designing and measuring strength properties

The proposed article is published before by the same authors. You may check the below links:

https://www.tandfonline.com/doi/abs/10.1080/10803548.2020.1796295

DOI:  https://doi.org/10.1080/10803548.2020.1796295

Kropidłowska, P., Irzmańska, E., Zgórniak, P., & Byczkowska, P. (2021). Evaluation of the mechanical strength and protective properties of polycarbonate toecaps subjected to repeated impacts simulating workplace conditions. International Journal of Occupational Safety and Ergonomics27(3), 698-707.

So, having reviewed the article, I recommend the article must be strongly rejected, the authors should not do this aging and they must abide by the research ethics of publication.

Author Response

Reviewer #2:

The proposed article is published before by the same authors. You may check the below links:

https://www.tandfonline.com/doi/abs/10.1080/10803548.2020.1796295

DOI:  https://doi.org/10.1080/10803548.2020.1796295

Kropidłowska, P., Irzmańska, E., Zgórniak, P., & Byczkowska, P. (2021). Evaluation of the mechanical strength and protective properties of polycarbonate toecaps subjected to repeated impacts simulating workplace conditions. International Journal of Occupational Safety and Ergonomics27(3), 698-707.

So, having reviewed the article, I recommend the article must be strongly rejected, the authors should not do this aging and they must abide by the research ethics of publication.

We strongly disagree with this allegation. As a proof of my words, I enclose below scans from the anti-plagiarism program which illustrate the fact that two articles mentioned are not identical and have nothing to do with each other. The anti-plagiarism system rated the article similarity at 8% of duplicate text including the publisher's standard transcripts, as can be seen below.

 It is worth emphasizing that the reviewer states that they are similar, quote: ”article is published before by the same authors”, but does not indicate any evidence in this regard. For a full explanation, I present a substantive explanation to confirm that the article does not contain similarities with the indicated article by the Reviewer.

I would like to explain that the article concerns the aspect related to the material design of toecaps. The publication describes the use of the finite element analysis method for impact testing in order to enable future footwear manufacturers to perform an initial assessment of the application of the toecap. An example of the application of the finite element method can be used by designers of safety footwear to optimize the spatial geometry of the toecap and to choose the appropriate construction material. Reviewer accused the Authors of the lack of research ethics of publication because, according to him, the article from 2020 entitled: “Evaluation of the mechanical strength and protective properties of polycarbonate toecaps subjected to repeated impacts simulating workplace conditions” published in the International Journal of Occupational Safety and Ergonomics (JOSE) (https://www.tandfonline.com/doi/full/10.1080/10803548.2020.1796295 ) is identical and suggests an association of self-plagiarism.

The article: Evaluation of the mechanical strength and protective properties of polycarbonate toecaps subjected to repeated impacts simulating workplace conditions, published in the International Journal of Occupational Safety and Ergonomics, does not apply to finite element analysis and does not address the issue of numerical simulations or modeling in any respect.

The publication includes laboratory tests with the use of research equipment in the form of an impact tester. The strokes were repeated until the toecaps were defragmented. This was done in order to verify how many strokes a commercially available toecap can withstand. In the work environment, employees are exposed to multiple hits in the area of ​​the toes, hence the purpose of performing multiple strikes of the toecaps was to simulate the actual conditions of footwear use in the work environment. Research effect were expressed as the height of toecap clearance, which has a direct bearing on the safe use of protective footwear. Additionally, external and internal sides of toecaps were scanned in three dimensions after each impact and reverse engineering was used to analyze deformations in toecap geometry by comparing the shape of the toecaps before and after impact. Three-dimensional scanning made it possible to measure the remaining safe distance for toes between the footwear sole and the impacted toecap surface.

The article shows that, during use the protective performance of the toecaps may be compromised already after a few impacts and the article entitled Evaluation of the mechanical strength and protective properties of polycarbonate toecaps subjected to repeated impacts simulating workplace conditions         does not repeat the results presented in Manuscript ID: : ijerph-1878045.

Reviewer 3 Report

Thank you for your work on this study. This work is a welcome addition to the body of knowledge in this area. Please find below some comments and suggestions that I hope will prove helpful. (No file uploaded - all comments within this text.)

Line 39 - do you mean the application of these materials for toecaps or just more widely in industry? (It wasn't entirely clear)

Line 46 - "other toecap materials" - should this be specifically metallic? Composite toecaps would require a totally different approach.

Lines 59-64 - it would be worth mentioning that one standard is an "in-shoe" test and one is for the toecap alone for clarity and that you are basing it on this. [as a side note, do the authors think that these two tests are comparable in reality? Simulations would potentially answer this question too!]

Line 70 - an explanatory figure would be useful here (maybe an adaption of the figure in the EN standard - copyright permitting)

Line 109 - table 1 - is the mean thickness measured from the actual toecaps or from the model? What size were the toecaps?

Line 116 - this is a common problem due to the crudeness of the toecaps - I appreciate your difficulty here

Line 155 - Figure 2 has gained some formatting errors in compilation.

Line 155 - should Figure 2 be at the start of section 2, with subsections A-E following accordingly to step through the procedure?

Line 188 - I think this should be section 4? And maybe "Results and Analysis"

Line 242 - could the text from here be a "Discussion" section?

The conclusions should remind the reader that the crush test must also be passed so this study covers only one element of the testing regime.

In reality, these toe caps are crudely produced and can be inconsistent in terms of wall thickness and flatness of contact surface with the sole of the shoe. Same batch samples can vary widely. It would be good to relate this study back the the practicalities; the manufacturing limits for toe caps based on the outcome of any simulations must be considered. However, it is fully appreciated that such simulations may permit the use of novel materials that may not otherwise be considered.

And finally, the aim of the study seems to be focused around optimising materials and potentially designs in order to be able to pass reach the regulatory standards; but design may also be hindered by the standards (e.g. a lattice structure may prove structurally and weight efficient but would not be permitted under the current standards). The authors might consider recognising that this study is underpinned by meeting these standards that may in fact be outdated and prohibitive (though this may not be their opinion so I will leave it with them to decide!).

Author Response

Reviewer #3:

Thank you for your review and comments. Changes have been made in the body of the text.

Below please find our specific responses.

Line 39 - do you mean the application of these materials for toecaps or just more widely in industry? (It wasn't entirely clear)

Clarified. It concerns the use of materials in toecaps.

Line 46 - "other toecap materials" - should this be specifically metallic? Composite toecaps would require a totally different approach.

Corrected.

Lines 59-64 - it would be worth mentioning that one standard is an "in-shoe" test and one is for the toecap alone for clarity and that you are basing it on this. [as a side note, do the authors think that these two tests are comparable in reality? Simulations would potentially answer this question too!]

We would like to thank the Reviewer for this comment. Tests perform according to EN ISO 20344 and EN 12568 are comparable because both tests are designed to verify protective properties of toecaps and assess safe post-impact clearance.

Line 70 - an explanatory figure would be useful here (maybe an adaption of the figure in the EN standard - copyright permitting)

This is illustrated in Figures 3 and 4.

Line 109 - table 1 - is the mean thickness measured from the actual toecaps or from the model? What size were the toecaps?

The simulations used 3d scans of real toecaps. The thickness was measured for actual toecaps. The size of the tested toecaps was 9.

Line 116 - this is a common problem due to the crudeness of the toecaps - I appreciate your difficulty here

Thank you for your comment.

Line 155 - Figure 2 has gained some formatting errors in compilation.

It was corrected.

Line 155 - should Figure 2 be at the start of section 2, with subsections A-E following accordingly to step through the procedure?

Thank you for this suggestion. It was corrected.

Line 188 - I think this should be section 4? And maybe "Results and Analysis"

It was corrected.

Line 242 - could the text from here be a "Discussion" section?

Thank you for this suggestion. It was corrected.

The conclusions should remind the reader that the crush test must also be passed so this study covers only one element of the testing regime.

In reality, these toe caps are crudely produced and can be inconsistent in terms of wall thickness and flatness of contact surface with the sole of the shoe. Same batch samples can vary widely. It would be good to relate this study back the the practicalities; the manufacturing limits for toe caps based on the outcome of any simulations must be considered. However, it is fully appreciated that such simulations may permit the use of novel materials that may not otherwise be considered.

And finally, the aim of the study seems to be focused around optimising materials and potentially designs in order to be able to pass reach the regulatory standards; but design may also be hindered by the standards (e.g. a lattice structure may prove structurally and weight efficient but would not be permitted under the current standards). The authors might consider recognising that this study is underpinned by meeting these standards that may in fact be outdated and prohibitive (though this may not be their opinion so I will leave it with them to decide!).

We would like to thank the Reviewer for this comments. It partially corresponds with the comment of the Reviewer 1. Changes were made to the abstract and conclusion.

Round 2

Reviewer 1 Report

This paper simply says that the authors were able to simulate tip toe stress and deformation using ANSYS software. A SIMPLE USE OF COMMERCIAL SOFTWARE DOES NOT FIT EVEN FOR A USER CONFERENCE PAPER OF EMPLOYED SOFTWARE. Furthermore, they failed to correctly simulate an impact event which is dynamic. I find nothing new to the knowledge society from this paper, and thus I cannot recommend publication of it in a journal. 

Furthermore, several points are disagreeable:

L21-23: The presented numerical tool for impact simulation may be used by the designers of safety footwear to optimize the spatial geometry of toecaps and to verify the construction of toecaps.

L309: The objective of the study was to develop a numerical tool for toecap strength testing.

=> What numerical tool was presented? Anybody trained in FE analysis can set up their tip toe FE model without this paper. People does not receive any aid from this paper about numerical tool.

L113-116: The presented methodology can be used to support the process of production preparation and considerably enhance impact resistance analysis as it can predict critical stress 114 and strain conditions leading to toecap failure, which would translate into hazardous toe- 115 cap wall deformations during footwear use.

=> It is vague self-compliment for Fig. 1 (research procedure) without any logical background. Fig. 1 (research procedure)  is only a routine procedure in reviewer’s standard.

Section 2.2 on Johnson-Cook Constitutive model

=> The author’s understanding of the strain-rate dependent constitutive model leaves greatly to be desired. Indeed L157-165 does not fit for a journal-level paper.

L252-254: The available literature predominantly focuses on the STATIC ASPECTS of protective product testing, with the evaluated factors being correlations with absorption energy, mean crushing load, maximum peak load, stress distribution, and toecap deformation 254 [5,12,31,32].

The current paper is nothing to do with Dynamic Aspects.

Section 5. Discussions => The authors are not discussing what was found in this paper. They are narrating literature review.

L260-262: The tested S3F3 and S3F3b toecaps differed in their geometry and analyses took into account nonlinear material hardening upon impact described by the Johnson–Cook model. The JC model was not designed for non-linear analysis. Any constitutive model describes work hardening behavior in flow regime.

=> What numerical tool was presented? Anybody trained in FE analysis can set up their tip toe FE model without this paper. People does not receive any aid from this paper about numerical tool.

L313-313: The developed numerical model simulating impact testing makes it possible to conduct preliminary evaluation without purchasing the necessary materials and launching production.

=> This paper is not doing impact testing. It simulates quasistatic phenomenon because the constitutive model was set as such. What numerical model was developed? Anybody trained in FE analysis can set up their tip toe FE model without this paper. People does not receive any aid from this paper about numerical tool.

ð  

ð L314-316: This study presented results on the dynamic response of safety toecap models made of various steels and lexan. The undertaken work contributes to better understand the structural impact behavior of toe cap models.

ð This paper is nothing to do with Dynamic Response.

L316: Also it is useful for people in that this paper prevent from wrong direction in toecap design.

ð  

This paper does not prevent people from wrong direction. This paper does not correct at all about the knowledge of the people. Please note that nothing in this paper is new to the knowledge society.

Reviewer 2 Report

 The revised manuscript can be accepted for publication.